# Vitamin D and the Central Nervous System: Causative and Preventative Mechanisms in Brain Disorders

**DOI:** 10.3390/nu14204353

**Published:** 2022-10-17

**Authors:** Xiaoying Cui, Darryl W. Eyles

**Affiliations:** 1Queensland Centre for Mental Health Research, The Park Centre for Mental Health, Wacol Q4076, Australia; 2Queensland Brain Institute, University of Queensland, St Lucia Q4076, Australia

**Keywords:** brain, development, vitamin D deficiency, neuroprotection, disease mechanisms

## Abstract

Twenty of the last one hundred years of vitamin D research have involved investigations of the brain as a target organ for this hormone. Our group was one of the first to investigate brain outcomes resulting from primarily restricting dietary vitamin D during brain development. With the advent of new molecular and neurochemical techniques in neuroscience, there has been increasing interest in the potential neuroprotective actions of vitamin D in response to a variety of adverse exposures and how this hormone could affect brain development and function. Rather than provide an exhaustive summary of this data and a listing of neurological or psychiatric conditions that vitamin D deficiency has been associated with, here, we provide an update on the actions of this vitamin in the brain and cellular processes vitamin D may be targeting in psychiatry and neurology.

## 1. Introduction

The vitamin D receptor is part of the nuclear receptor super family containing members such as testosterone, estradiol, cortisol, progesterone, vitamin A derivative all-trans retinoic acid and the thyroid hormones [1]. These factors have important roles in the differentiation of all organs including the brain. It is now 21 years since we first suggested that vitamin D was a “possible” neurosteroid [2]. At that time, all we knew from existing research was that immunohistochemical evidences for the vitamin D receptor (VDR) had been shown in non-neuronal glial cells and that vitamin D may affect neurotrophic factor expression in vitro. There was very little knowledge about how vitamin D could exert its genomic effects in brain cells and a complete absence of any understanding about whether, like other neurosteroids, vitamin D would also have more rapid non-genomic actions. Over the last two decades, research on neurons, non-neuronal brain cells, in both human and animal brains, has confirmed not only does the brain possess the molecular machinery for vitamin D’s actions, but also that VDRs are functional, that liganded VDRs directly regulate the expression of target genes and that vitamin D in the form of 1,25-hydroxyvitamin D_3_, 1,25(OH)_2_D_3_ can rapidly alter ion channel function [3]. Moreover, only very recently have new epigenetic mechanisms been revealed as gene regulatory pathways that vitamin D targets to affect gene expression in the developing brain [4,5]. As a result of all this research, vitamin D can now be considered an important steroid in brain development [6]. More recently the neuroprotective functions of vitamin D have been highlighted in models of adult brain disorders. Here we integrate these data and outline the molecules and processes vitamin D appears to target in both developing and mature brains and how such actions shape behaviour and brain function.

We are now aware of the large number of non-skeletal targets for vitamin D. In brain cells, changing vitamin D status alters cytokine regulation and has been shown to affect cell differentiation, neurotrophin expression, intracellular calcium signalling, neurotransmitter release, anti-oxidant activity, anti-inflammatory actions, stress responsivity and the expression of genes/proteins important to neuron physiology [7,8]. The purpose of this short update is not to exhaustively summarise all such associations, but to bring the reader up to speed with recent findings using the very latest techniques in molecular manipulation/quantitation and cell visualisation.

With respect to brain disorders, there is now reliable epidemiology linking embryonic or neonatal vitamin D deficiency with an increased risk of neurodevelopmental disorders, such as schizophrenia [9,10], autism [11,12,13,14] and, more recently, attention deficit hyperactivity disorder (ADHD) [15,16,17]. There are also numerous studies suggesting adult vitamin D deficiency can be correlated with certain neurodegenerative conditions. This clinical epidemiological literature has been reviewed elsewhere [18,19], and we will return to this in the final section of this article. Our purpose here will be to focus on the latest preclinical studies modelling these epidemiological links and to discuss plausible biological mechanisms behind disease-relevant phenotypes.

Although we were pioneers in this area, works from a number of laboratories have firmly established the biological plausibility for how low levels of vitamin D could adversely affect how brains form and how this could lead to subtle changes in brain function and behaviour. Our task now is to discover exactly how low levels of vitamin D impair the function of specific brain cells/circuits leading to adult brain disorders and whether correcting vitamin levels and/or the processes affected by impaired vitamin D signalling can diminish phenotype/symptom severity.

## 2. Vitamin D Signalling in the Brain, the Basics and Controversies

Early studies reporting levels of vitamin D metabolite levels in the brain produced widely varied results. This has been to a large extent due to technical difficulties in extraction and quantification methods. The major circulatory form of vitamin D, 25-hydroxyvitamin D_3_, (25(OH)D_3_), and its active hormonal form, 1,25-hydroxyvitamin D_3_, 1,25(OH)_2_D_3_, have been reported to be present in brain [20,21]. Whilst the exact concentrations are debatable, they are routinely reported as far lower than blood levels. The later use of LC/MS/MS with isotope dilution provides absolute chemical identification due to its selectivity and greater sensitivity. Using this technology, one group shows 25(OH)D_3_ to be around (4 ng/g tissue) in rodent brain [22,23]. This group used atmospheric pressure, photoionisation which may avoid ion-suppression artifacts which may have affected earlier methods. Very recently, Fu and colleagues developed a method to measure vitamin D metabolites in the human brain and found 25(OH)D_3_ was measurable, but levels were far lower than those found in the blood around 0.2–0.3 ng/g tissue [24].

With respect to the VDR, despite immunohistochemical studies confirming its presence in human, chick, rat, mouse and zebrafish brains, [25,26,27,28,29,30] claims that some of the antibodies used may have been less-than-specific and have invited dispute [31,32]. Now unambiguous evidence has been provided using mass-spectrophotometry to electrophoretically resolve proteins from adult rodent brains to identify five unique VDR peptides with a confidence interval greater than 99% [33]. There is also close anatomical overlap between brain regions in rats and humans, with respect to VDR location [25,26,27,29,34,35]. Consistent close cross-species overlap in VDR distribution validates the use of rodents in modelling vitamin D-related brain outcomes.

Immunohistochemistry, mRNA and protein analyses all reveal a gradual expression of the VDR in the developing brain [33,36,37,38]. In the developing brain, the VDR is concentrated in differentiating fields, such as the ependymal surface of the lateral ventricles [39] (the greatest source of cell division in the brain), consistent with vitamin D actions as a differentiation agent. In particular, VDR expression in the nucleus of dopamine neurons which appear very early in brain development correlates tightly with the ontogeny of these neurons [35]. New genomic technologies have now allowed VDR location and activity to be assessed in animal brains. Using clustered regularly interspaced short palindromic repeats (CRISPR), Liu et al. inserted a Cre-expression sequence driven by the endogenous VDR promotor. When such animals were mated with a tdTomato reporter mouse ((Ai14; B6.Cg-Gt(ROSA)26Sor tm14 (CAG-tdTomato)Hze/J), intense tdTomato fluorescence was detected in multiple brain regions, including caudate putamen, amygdala, reticular thalamic nucleus, cortex and less intense tdTomato fluorescence in hippocampus, hypothalamus, paraventricular thalamic nucleus (PVH), dorsal raphe and bed nucleus of the stria terminalis (BNST). RNAscope confirmed the presence of VDR RNA in these same tdTomato positive neurons. However, perhaps most importantly, 1,25(OH)_2_D_3_ (5 μM) selectively depolarised the tdTomato positive but not negative neurons in the PVH, indicating vitamin D directly regulates neuronal activity, but only in VDR-expressing neurons [40].

Immunohistochemical evidences for the enzymes that convert 25(OH)D_3_ to the active hormonal form of vitamin D, 1,25(OH)_2_D_3_, *CYP*27B1, and enzymes responsible for its breakdown, *CYP*24A1, have been provided in the foetal [41] and adult human brain [25,42], suggesting the active hormone can be made or removed locally in human brains. The most thorough investigation of expression of the major vitamin D metabolising enzymes *CYP*24A1, *CYP*271B and *CYP*271A in brain cell types was undertaken by Landel and colleagues [43]. Using primary cultures of neurons, astrocytes, microglia and oligodendrocytes (encompassing all major brain cell types), this group showed a broad distribution across all cell types but at a lower level than the kidney and liver. Strikingly, however, the addition of 1,25(OH)_2_D_3_ induced a profound upregulation of *CYP*24A1 only in astrocytes and microglia, a finding also previously observed [44]. Again, genomic technologies have been employed to study the activity of these enzymes. Hendrix et al. created a transgenic mouse that expresses the luciferase reporter under the control of a full-length human CYP27B1 promoter. The organ distribution of CYP27B1 luciferase activity is similar to endogenous CYP27B1 with highest activity in the kidney, testis, brain, skin and bone [45]. Unfortunately, this useful model apparently has not yet been utilised to study what factors upregulate CYP27B1 brain activity.

## 3. Vitamin D and Normal Brain Development

A wealth of experimental evidence exists regarding the plausibility for how altering vitamin D signalling could affect critical events in brain development, such as axonal elongation, neurotransmitter synthesis, neurotrophin production and later brain function. Although there are some comprehensive reviews in this space [46,47], here, we bring this up to date.

### 3.1. Vitamin D and Neurite Growth

The addition of 1,25(OH)_2_D_3_ to embryonic hippocampal neurons in culture increases neurite outgrowth possibly via an increased nerve growth factor (NGF) [48,49]. We also now have new unpublished data replicating the neurite-promoting potential of 1,25(OH)_2_D_3_ in developing dopamine neurons differentiated from (a) a neuroblastoma cell line and (b) dopamine neurons in explant mesencephalic cultures. This action is in line with most other neurosteroids. Now, we are exploring the molecular mechanisms behind these effects. Others have chosen to lesion peripheral neurons and to show that vitamin D enhances axonal repair, myelination and functional recovery [50,51]. The 1,25(OH)_2_D_3_ treatment also restores neurite outgrowth in models where it is impeded. The knockdown of the important neuronal epigenetic regulator, MeCP2, in cortical neurons blunts neurite extension potentially through reducing the activity of NFκB pathways. The 1,25(OH)_2_D_3_ restores normal outgrowth in this model [52]. Vitamin D deficiency has also been shown to reduce peripheral nerve fibre density [53], and vitamin D has been considered a potential therapeutic in spinal cord repair [54].

### 3.2. Vitamin D and Neurotrophic Factors

The first evidence that vitamin D had any action on brain cells came from the early studies from Didier Wion’s group in glioblastoma cells. Early studies showed that vitamin D promoted the expression of NT-3, NT-4 and nerve growth factor (NGF) [48,49,55,56,57]. Vitamin D-mediated increases in NGF were shown to be highly relevant to neuronal survival in vitro [48,58,59,60].

Given its prominent role in dopaminergic neuron differentiation (see below) and survival in earlier studies, there was a strong focus on vitamin D and neurotrophic factors important for dopaminergic neurons, such as glial cell line-derived neurotrophic factor (GDNF) [61,62] and now brain-derived neurotrophic factor (BDNF) [63]. Blocking vitamin D-mediated increase in GDNF synthesis prevents vitamin D’s trophic effects on these neurons [64]. We have described the genomic proof that vitamin D regulates the transcription of both receptors for GDNF. The 1,25(OH)_2_D_3_ suppresses GDNF family receptor alpha 1 (GFRa1), but ligand bound VDR binds to the promoter of the other major receptor for this neurotrophin, the proto-oncogene tyrosine-protein kinase receptor Ret (C-Ret), to upregulate C-Ret expression. Accordingly, the maternal absence of vitamin D decreases C-Ret expression in the developing rat mesencephalon [65].

The effect of vitamin D on neurotrophic factors in the developing brain has been far less explored. One study showed DVD-deficiency in rats reduced NGF and GDNF protein in neonatal brains [66]. Reductions in BDNF and transforming growth factor-β1 (Tgf-β1) in DVD-deficient embryonic mouse brains have also been described [67].

Most recently, there have been numerous studies showing a convincing link between hippocampal BDNF levels and vitamin D status in adult animals. Consistent with findings in development where the absence of vitamin D leads to a BDNF deficit, very recent studies show chronic treatment with 1,25(OH)_2_D_3_ induces a profound upregulation in BDNF in rat hippocampus [68]. Others show stress- or drug-induced memory deficits correspond with decreased hippocampal BDNF expression that can be restored via chronic supplementation with high doses of cholecalciferol [69,70]. Another study revealed age-induced memory deficits could be reversed with cholecalciferol supplementation with corresponding increases in hippocampal BDNF and NGF expression [71]. In a model of type 2 diabetes in mice, BDNF and phosphorylation of its downstream effector CREB are reduced in the brain, and cholecalciferol restores these and associated behavioural deficits [72].

In summarising this data, 1,25(OH)_2_D_3_
*increases*, and the developmental dietary absence of vitamin D *reduces* the expression of these crucial neurotrophic factors in neurons [73] and glia in developing and adult brains. Vitamin D’s regulation of neurotrophic factors remains a central feature in brain ontogeny.

### 3.3. Vitamin D Regulates the Development of Dopamine Neurons

Since we first reported the VDR within the human substantia nigra [25], we have gone on to confirm the VDR is prominent in neuromelanin containing Tyrosine Hydroxylase (TH) positive human nigral neurons [35]. Vitamin D deficiency during development leads to a reduction in specification factors crucial for dopamine neurons [74,75], alters dopamine turnover [76] and leads to decreased lateral positioning of dopamine neurons that will form the substantia nigra [75]. Importantly, similar developmental anomalies in dopamine neuron ontogeny created by other adverse developmental exposures are rescued by maternal treatment with 1,25(OH)_2_D_3_ [77]. To better understand the molecular processes involved, we have used neuroblast cells in which the VDR has been over-expressed. In such a model, we have shown 1,25(OH)_2_D_3_ increases TH mRNA [78], drives cells down a dopaminergic lineage [79] and directly targets the promoter of Catechol-o-methyl transferase (COMT), an important enzyme in brain dopamine turnover [79].

One very recent study has examined microRNAs in developing dopamine neurons as epigenetic factors in neuronal maturation. This study showed that in flow cytometry-sorted E14 dopamine neurons from DVD-deficient rat embryos, the expression of a number of microRNAs was increased and that gene-ontology analysis indicated these microRNAs were involved in neuronal differentiation. These authors then examined the functional consequences of this in developing dopamine neurons and showed that some of these microRNAs decreased neurite outgrowth [4]. This represents the first evidence in the brain indicating how vitamin D may epigenetically differentiate dopamine neurons.

Taken together, the decreased expression of dopamine-promoting neurotrophic and specification agents and delayed dopamine neuron maturation in DVD-deficient embryonic brains along with the abundant in vitro evidences for 1,25(OH)_2_D_3_ as a maturation agent for dopamine neurons, all confirm a central role for vitamin D in dopamine ontogeny. Given the links between DVD-deficiency and schizophrenia [9,10] and that dopamine abnormalities are perhaps causal in this disease, further studies examining this possible mechanistic link are needed.

## 4. Developmental Vitamin D Deficiency Effect on Brain Function and Behaviour

The long-term effects of DVD-deficiency on offspring behaviour have also been extensively studied in both rats and mice. Outcomes vary based on the species used/strain and duration and degree of vitamin D deficiency. Notably, there are alterations in critical early dam/pup interactions, such as maternal licking and grooming [80] and pup ultrasonic vocalisations [5,80], along with signs of delayed motor development [5]. As juveniles or adults, offspring have social behavioural deficits and alterations in stereotyped behavioural phenotypes of relevance to autism [5,80]. Interestingly, vitamin D deficiency leads to subtle increases in testosterone levels in maternal blood [81], a finding which has long-been considered a risk-modifying factor in autism [82]. Additionally, foetal male brains from vitamin D deficient dams have increased testosterone compared to control males. This study went on to show silencing of the major enzyme involved in testosterone breakdown, aromatase, by hypermethylation of its promoter in the brain may be the mechanism [81].

DVD-deficiency also leads to long-term changes in adult behaviours. Novelty [83], or exposure to psychomimetics, such as the N-methyl-D-aspartic acid receptor (NMDA-R) antagonist, MK-801 [84,85,86] or amphetamine [87], all increase locomotor activity in the adult offspring of DVD-deficient dams. Along with sensitivity to these exposures/agents that release dopamine, DVD-deficient rats have a greater response to antipsychotics that block dopamine 2 receptors [84,88]. Brain functional measures are also abnormal in these animals. Long-term potentiation, a cellular correlate of learning and memory and latent inhibition, a measure of attentional processing are also abnormal in DVD-deficient adults [89,90]. With respect to cognition, DVD-deficient offspring have impaired response inhibition, a deficit normalised by the antipsychotic clozapine [91]. Associative learning is also impaired in DVD-deficient adults [92].

Only recently have investigators begun to assess the impact of vitamin D treatment in dams exposed to either DVD-deficiency or another developmental animal model that induces brain changes, maternal immune activation. Such interventions reverse behavioural phenotypes produced by these models related to anxiety, depression cognition, stereotyped and social behaviours [93,94].

As discussed previously, given the stark differences in the effects of DVD-deficiency on brain development between rats and mice, it is not surprising that the behavioural phenotypes in adult offspring also vary widely [95]. For instance, although there are some similarities in novelty-induced hyperlocomotion [84] and perseverative responses [96], outcomes vary widely in tests of exploratory behaviour [88,97] and locomotor response to psychomimetics [98].

Whether DVD-deficiency affects cognitive performance in children is not clear. Two studies show DVD-deficient children have delayed cognitive development [99,100]. However, these findings were not replicated in much larger studies [15]. A recent meta-analysis that summarised 31 studies could find no conclusive evidence for the association between maternal or child vitamin D status and behavioural or cognitive outcomes in children and adolescents. A summary of twelve mother–child studies (n = 17,136) and five studies just in children (n = 1091) showed low maternal or child 25(OH)D_3_ levels led to impaired behavioural outcomes in children. In contrast, fifteen mother–child studies (n = 20,778) and eight studies in children (n = 7496) showed no association [101]. A large Finnish nested case-control study (1607 children with learning difficulties with matched controls) also showed no link between maternal 25(OH)D_3_ levels and behaviour [102].

One small randomised clinical trial (55 infants) in healthy term infants showed a low dose of vitamin D supplementation might benefit gross motor function compared to a higher dose of vitamin D supplementation, but this study lacked a placebo control [103]. There is also a not-insubstantial number of studies that link maternal or childhood vitamin D deficiency to neurodevelopmental psychiatric disorders such as autism, ADHD or schizophrenia [9,10,11,13,14,100,104,105]. This association is not universally replicated [106,107]; however, the mean population 25(OH)D_3_ levels in these later studies were much higher, making the association difficult to test. Randomised controlled trials of cholecalciferol supplementation in children with ASD or ADHD show diverse outcomes, some showed no beneficial effects and others reported some symptom relief [108,109,110,111,112,113,114]. In the future, trials examining vitamin D levels in mothers/children prior to supplementation are required as supplementing vitamin D-sufficient mothers or children is likely to have minimal effect, as shown in other clinical trials with vitamin D [115].

## 5. Vitamin D Is Neuroprotective in Neurons and Adult Brain

There are now sufficient epidemiological studies also linking low levels of 25OHD_3_ with various neurodegenerative conditions such as Alzheimer’s disease, Parkinson’s disease and multiple sclerosis. Though many such studies suffer from reverse causality (i.e., low levels of vitamin D are a consequence of disease-induced behavioural changes rather than causal), there has been sufficient interest to launch numerous clinical supplementation trials [116,117,118]. There have also been studies linking exposure to various toxins with low 25OHD_3_ levels. Such links have stimulated researchers to initiate mechanistic studies in neuronal cell systems and animal models. Along with the wealth of studies showing vitamin D is neuroprotective via neurotrophin production, here, we have chosen not to focus on the clinical epidemiology, but rather on vitamin D’s regulatory actions on calcium in the brain and its neuroprotective actions against reactive oxygen species (ROS) and inflammation as well as its actions in mitigating stress as plausible prophylactic/therapeutic mechanisms. We summarised the potential mechanisms underlying the action of vitamin D (Figure 1). Much of this work is very recent.

*Calcium-related mechanisms (in blue). Non-genomic actions*. The 1,25(OH)_2_D_3_ acutely facilitates calcium influx via L-VGCC. The 1,25(OH)_2_D_3_; can also directly bind to TRPV1 to induce calcium influx. Both actions facilitate neuronal function. *Genomic actions.* Chronic vitamin D treatment decreases Cav1 (a L-VGCC subunit) expression which will reduce calcium influx in response to ROS/inflammation or stress.

*ROS/inflammation-related mechanisms (in purple)*. ROS enhances NF-κB nuclear translocation to promote proinflammatory cytokine production. The 1,25(OH)_2_D_3_ acts to inhibit this nuclear translocation. By also inhibiting NF-kB expression, 1,25(OH)_2_D_3_ reduces iNOS expression and thus, reduces NO production. Finally, 1,25(OH)_2_D_3_ increases Nrf2 to enhance the expression of anti-oxidant enzymes including GP, CAT, SOD and HO-1, thus countering ROS toxicity. The 1,25(OH)_2_D_3_ also directly facilitates Nrf2 nuclear translocation.

*Stress-related mechanisms (in green)*. Stress increases corticosterone synthesis. In the brain, corticosterone increases inflammatory cytokine production via its receptor (GR). The 1,25(OH)_2_D_3_ is neuroprotective by antagonising GR expression.

*Neurotrophin-related mechanisms (in yellow)*. A variety of neurotrophic factors have been shown to counter the effects of stress- or toxin-induced neuronal damage. Historically, 1,25(OH)_2_D_3_ has been shown to increase NGF, GDNF, BDNF, NT-3 and NT-4 under such conditions.

### 5.1. Calcium Regulation

Calcium transients are required for normal neuronal function, but if calcium is unbuffered, it is toxic to brain cells. For more than 30 years, we have known how vitamin D regulates calcium uptake in bone cells [119,120] and similar mechanisms appear to act in neurons and the brain. In cultured neurons, 1,25(OH)_2_D_3_ retards calcium influx via the downregulation of L-type voltage-sensitive calcium channels, thus potentially preventing toxic outcomes [59,121,122,123]. In contrast, the rapid non-genomic actions of vitamin D produce the opposite effect, increasing calcium influx in cortical slices, a process that is again dependent on L-type calcium channels [124]. In a seminal study, the non-genomic rapid actions of 1,25(OH)_2_D_3_ were investigated in cortical neurons using calcium imaging, electrophysiology and molecular biological techniques. This study confirmed that physiological concentrations of 1,25(OH)_2_D_3_ lead to rapid calcium influx, but only in some neurons. Somatic nucleated patch recordings revealed a rapid, 1,25(OH)_2_D_3_-evoked increase in high-voltage-activated calcium currents mediated by L-type voltage-gated calcium channels [125]. Whether any of these actions are caused by the putative membrane VDR, Protein disulfide-isomerase A3 (PDIA3), in the brain remains unknown. Genetic variants in L-type voltage-gated calcium channels continue to be implicated in schizophrenia [126]. Given the epidemiological links between DVD-deficiency and schizophrenia [9,10], continued studies in this area are warranted.

Recent research also discovered that 25OHD_3_ and 1,25(OH)_2_D_3_ directly bind to the transient receptor potential vanilloid subfamily member 1 (TRPV1) channel [127]. Binding to the same region as the TRPV1 agonist capsaicin, 25OHD_3_ can weakly activate TRPV1 and inhibit capsaicin-induced TRPV1 activity. TRVP1 activity modulates immune cell activation and cytokine production through regulating intracellular calcium to mediate nociceptive signals. This, therefore, may be one mechanism for how vitamin D may modulate nociceptive pain pathways, as vitamin D deficiency has been linked to chronic pain [128], although oxidative mechanisms have also been proposed [129].

### 5.2. ROS and Inflammation

Vitamin D increases anti-oxidants, such as glutathione and cytochrome c, to mediate anti-oxidant actions in cultured neurons [121,130] and the brain [131,132]. Along with extracellular calcium, vitamin D deficiency in the adult brain increases ROS along with producing impairments in gamma-aminobutyric acid (GABA) and glutamate release. Importantly, reintroducing dietary vitamin D normalises all deficits [133].

In the brain, microglia are the immunologically responsive cells responsible for the production of inflammatory regulators such as nitric oxide (NO). Early studies showed 1,25(OH)_2_D_3_ blocks inducible nitric oxide synthetase in the rat brain in response to autoimmune or inflammatory factors [134,135,136]. Later studies suggested that oxidative stress may upregulate *CYP*27B1 in microglia to induce 1,25(OH)_2_D_3_ production locally at the site of NO or ROS production to mediate vitamin D’s anti-oxidant effects in the brain [133,137,138].

In primary cultured neurons, hypoxia induces apoptotic cell death and interferes with normal calcium signalling. One study, that chose to use cholecalciferol rather than calcitriol, showed that when added to cultured primary neurons exposed to hypoxia, cholecalciferol (10 nM) was anti-apoptotic and preserved calcium signalling though this effect was reversed at high doses. Upregulation of hypoxia-induced factor (HIF)-1α and/or BDNF were considered as the possible protective mechanisms [139], though conversion to calcitriol was not assessed.

DVD-deficiency would also appear to render the foetal environment more prone to oxidative stress. Ali and colleagues reported that DVD-deficient rat placenta produces more of the inflammatory cytokines IL-6 and 1L-1β upon challenge with a viral inflammatory agent [140]. Separately, when microglia were cultured from DVD-deficient mouse brains, they were increased in number, were hyperproliferative and had increased ROS production. Culturing these cells in the presence of calcitriol reversed these changes [141].

In recent years, there appears to have been intense interest in vitamin D’s protective actions against ROS and inflammation in the brain induced by numerous disease models. For instance, in hypothyroid juvenile rats, cholecalciferol supplementation (100 or 500 IU/kg/day) prevented hypothyroidism-induced cognitive and learning memory impairments [142]. Plausible mechanisms included elevations in the anti-oxidant enzyme superoxide dismutase (SOD) and thiol content in hippocampus and cortical tissue and reductions in malondialdehyde (MDA), a marker of oxidative stress. Similar restorative outcomes on cognition have been achieved in models of acute inflammation and again these same anti-oxidant processes in the brain were invoked by vitamin D supplementation [143].

In spontaneously hypertensive (SH) rats, infusion of 1,25(OH)_2_D_3_ into the hypothalamic paraventricular nucleus (PVN), a brain region maintaining baroreflex and autonomic function, prevented the elevation of ROS stress-related proteins such as NOX2, NOX4 and p22^phox^. Chronic calcitriol infusion also attenuated microglial activation and reduced tumour necrosis factor (TNF)-α, IL-1β and IL-6 inflammatory cytokine production. Likely, mechanisms involved the inhibition of the high-mobility group box-1(HMGB1)-receptor for advanced glycation end products (RAGE)/Toll-like receptor (TLR4) and NF-κB in the PVN of SH rats [144].

The effect of vitamin D on NO is also detected in a rat model of cerebral ischemia reperfusion model. Seven days of calcitriol administration prior to ischemic surgery reduced stroke-induced elevation of MDA and NO and increased total anti-oxidant capacity. These effects could be attributed to vitamin D-increasing nuclear factor erythroid 2-related factor 2(Nrf2), a transcription factor that decreases oxidative stress following stroke and/or heme oxygenase (HO-1), a major cytoprotective enzyme with anti-oxidative, and anti-inflammatory properties [145,146].

In a traumatic brain injury model, calcitriol treatment reduced MDA production and promoted autophagic flux and activated Nrf2 pathways. Autophagy reduces oxidative stress by a timely removal of damaged substances [147]. Nrf2 is a redox-sensitive transcription factor that binds to anti-oxidant response elements to promote the expression of enzymes/proteins for detoxication and anti-oxidation [148]. Confirmation that these two processes were central to calcitriols protective actions in traumatic brain injury was shown by inhibiting the autophagy using chloroquine or deleting Nrf2 as either action blocked calcitriol’s protective effects [149].

In a model of lead-induced neurotoxicity and oxidative stress, decreases in anti-oxidant molecules GSH, SOD and catalase and increased ROS production are observed in rat cortex. Cholecalciferol reversed these changes possibly via Nrf2 and/or NF-κB mechanism [150]. In a model of experimentally induced epilepsy in young male rats, cholecalciferol not only reduced seizure severity, but corrected associated memory deficits, reduced extracellular calcium, restored anti-oxidative enzymes SOD and glutathione-related enzymes and reduced inflammatory cytokine production in the hippocampus [151].

One neurological disorder commonly associated with ROS-mediated brain damage is Parkinsons’s disease (PD). The motor deficits produced by PD are believed to be caused by dopamine neurons dying selectively in the substantia nigra and the associated reduced dopamine release in the dorsal striatum. This is modelled in animals via the intracranial delivery of relatively selective dopaminergic terminal toxins, such as 6-hydroxy dopamine (6-OHDA), or nigral toxins such as 1-methyl-4-phenyltetrahydropyrine (MPTP). In a 6-OHDA-treated mouse, cholecalciferol treatment two weeks after surgical lesion attenuated 6-OHDA-induced increases in the microglia marker CD11b, IL-1β and the oxidative stress marker p47phox, a primary modulator of NADPH oxidase activity and restored some motor functional deficits [152]. Another 6-OHDA study showed that cholecalciferol prevented characteristic losses in dopamine synthetic enzymes and transporters, preserved motor function as well as reduced lipid peroxidation [153]. Another study using MPTP showed co-administration with calcitriol improved motor deficits, reduced dopamine neuron toxicity and reduced ROS production [154]. The mechanism proposed was via an interaction between the liganded VDR and poly(ADP-ribose) polymerase-1(PARP1) to reduce its contribution to ROS-induced cell death. PARP1 pathways have been proposed as one mechanism for dopamine cell death in PD.

ROS production is also linked with other degenerative processes such as Alzheimer’s disease. Animal models for Alzheimer’s disease frequently employ genetic models that over-express the Tau or amyloid (Aβ) proteins that are closely linked with disease pathology. In an amyloid presenilin model, 13 weeks of vitamin D deficiency exacerbated the ROS production in this model by downregulating superoxide dismutase 1 (SOD1), glutathione peroxidase 4 and enhanced the expression of IL-1β, IL-6 and TNFα, along with increased Aβ production and Tau phosphorylation [155].

In another amyloid model where rats are injected with Aβ1-40, cholecalciferol supplementation reduced Aβ-induced MDA levels, increased SOD activity and improved hippocampal neuronal survival [156]. Prolonged vitamin D hypovitaminosis in mice (from 6-weeks-old to 6-months-old) altered the expression of genes involved in amyloid precursor protein homeostasis (Snca, Nep, Psmb5), oxidative stress (Park7), inflammation (Casp4), lipid metabolism (Abca1), signal transduction (Gnb5) and neurogenesis (Plat) [157]. In another distinct amyloid protein mutant mouse model (3xtg-AD), vitamin D levels are dramatically reduced at 9 and 12 months of age. Vitamin D supplementation in this model improves memory possibly via the suppression of collapsin response mediator protein-2 (CRMP2) phosphorylation [158]. An in vitro study has also shown 1,25(OH)_2_D_3_ alleviates Tau hyperphosphorylation and reduces ROS in a neuronal cell model treated with Aβ possibly via vitamin D’s role in enhancing GDNF [159].

Experimentally induced autoimmune encephalitis (EAE) is a mouse model mimicking the autoimmune reaction to myelin proteins inducing multiple sclerosis-like pathology. In this model, calcitriol administration decreased the severity of EAE by attenuating inflammation and demyelination at the spinal cord [160]. This same study also showed calcitriol treatment reduces lymphocyte, macrophage and activated microglia infiltration into the brain. EAE mice have increased blood brain barrier permeability thought to be due at least in part to a reduction in endothelial tight-junction proteins such as ZO-1. This same study showed calcitriol increases this proteins expression. Calcitriol also reduced EAE-induced lipid hydroxylation and enhanced the anti-oxidant enzymes glutathione peroxidase, catalase and SOD.

The 1,25(OH)_2_D_3_ not only suppresses inflammation in models of demyelination, but also enhances the differentiation or survival of oligodendrocyte progenitor cells in the spinal cord of a MOG35–55-induced EAE mouse model [161]. The promotion of oligodendrocyte maturation by 1,25(OH)_2_D_3_ is also observed in a cuprizone-induced EAE mouse model [162]. Krabbe disease is an inherited leukodystrophy. This demyelinating condition is caused by a galactocerebrosidase (GALC) deficit that results in loss of oligodendrocytes and demyelination. In an animal model of this disorder (GALC^twi/twi^; twitcher mouse), supplementing the heterozygous GALC^+/−^ dam from birth to weaning with cholecalciferol delayed onset of disease-induced locomotor deficits and tremors and extend the life span of offspring [163].

### 5.3. Glucocorticoids and Stress

Glucocorticoid release is the classic endocrine response to stress, and protracted exposure induces neuronal shrinkage then cell death [164]. The effects of 1,25(OH)_2_D_3_ and glucocorticoids in the body can be considered antagonistic [165,166,167,168,169]. Similarly, in the brain, 1,25(OH)_2_D_3_ antagonises the effects of dexamethasone (a corticosterone agonist) on hippocampal neuron differentiation and glucocorticoid receptor function [170]. Interestingly, dexamethasone can decrease 1,25(OH)_2_D_3_ synthesis in the hippocampus and prefrontal cortex, indicating this process is reversible [171].

Behaviourally, vitamin D antagonises the depression-like phenotypes induced by chronic cortisol administration in animals [172,173,174,175]. Possible mechanisms include regulation of hippocampal glucocorticoid receptors or the restoration of dopamine levels in the reward centres in the brain [175]. Chronic mild stress in rats leads to increased corticosteroids, inflammatory markers and decreased anti-oxidant enzymes SOD and glutathione peroxidase and catalase in the hippocampus and prefrontal cortex. Simultaneous cholecalciferol treatment reverses these stress-mediated effects [69,176].

Chronic unpredictable stress increases immobility in a widely used test of behavioural despair (tail suspension test). Cholecalciferol treatment reduces this stress-induced immobility. Inhibiting the synthesis of the neurotransmitter serotonin abolishes cholecalciferol’s actions in this test, suggesting vitamin D might act via increased serotonin synthesis [177]. Support for this idea comes from studies in glioblastoma cells showing functional VDREs are localised at −7 kb and −10 kb upstream of the serotonin, synthesising enzyme tryptophan hydroxylase [178]. Other studies chronically administering corticosterone replicate the cholecalciferol’s reversal of these stress-induced behaviours and suggest either alterations to glucocorticoid signalling in the brain or reductions in stress-associated ROS production in the brain are the protective mechanisms [172,179]. Stress can be measured in animals using other behavioural paradigms such as immobility in a forced swim test. Chronic unpredictable stress increases immobility in this test. Cholecalciferol supplementation reverses this immobility as well as reduces serum corticosterone/ACTH levels and increases BDNF and NT-3/NT-4 levels in the hippocampus [174].

DVD-deficiency may also alter maternal response to stress in rats [180] and mice [181] and can adversely affect maternal care [80], which is well-known to induce permanent changes in offspring stress-response [182]. The translational potential of these later findings is enhanced given that a randomised clinical trial showed that 50,000 IU of cholecalciferol alone every 2 weeks, or in combination with Omega 3 fatty acids (1 g/day), for 8 weeks significantly reduced anxiety and improved sleep quality in women of reproductive age with pre-diabetes and hypovitaminosis D [183].

## 6. Hypervitaminosis D and Adverse CNS Outcomes

While the role of hypovitaminosis D in brain function has been extensively investigated, the effect of hypervitaminosis D has received less attention. This is despite an inverted U-shaped cellular response to 1,25(OH)_2_D_3_ being frequently reported, i.e., both high and low levels of 1,25(OH)_2_D_3_ induce adverse outcomes. Hypervitaminosis D is rare in humans, generally resulting from excess vitamin D supplementation or diseases such as sarcoidosis that produces excess 1,25(OH)_2_D_3_ due to activated macrophages [184]. Though not focused on neurological conditions, an older review concluded “*There is accumulating evidence that both high and low serum calcidiol concentrations are associated with an increased risk of chronic diseases*” [185]. Hypervitaminosis D always results in hypercalcaemia, which may be toxic to brain function. Animal studies showed that high cholecalciferol intake, 25,000 IU/kg, for four consecutive days reduces brainwave activity [186]. Senescence-accelerated-mouse-phenotype (SAMP) strain-8 mice, an animal model of accelerated human ageing, showed a progressive increase in serum 25OHD_3_, which coexists with reduced cognitive function and increased capillary permeability [187]. Deleting Fibroblast growth factor (FGF-23) in mice increased *CYP*27B1 levels, leading to increased 1,25(OH)_2_D_3_ synthesis. FGF-23 null mice display early ageing, and deleting *CYP*27B1 delayed premature ageing [188]. Considering the risk of hypervitaminosis D, serum vitamin D levels should be monitored in clinical trials of vitamin D supplementation.

## 7. Conclusions and Future Challenges

Here we have provided an up-to-date summary of research detailing vitamin D turnover, synthesis genomic and non-genomic actions in the brain, neurons and non-neuronal cells. We advise caution when considering much of the prior literature employing either dietary restrictions which often produced hypo-calcaemia or the use of constitutive knock out models permanently reducing 1,25(OH)_2_D_3_ synthesis or impairing VDR signalling, as these models produce too many non-CNS effects to make brain-related outcomes interpretable. Although there are still strain and species differences when choosing a model organism, contemporary DVD- or AVD-deficient models in rodents do not produce hypocalcaemic offspring and continue to produce findings of apparent relevance to the fields of psychiatry and neurology.

Increasingly, epidemiological studies associate low levels of vitamin D either prenatally or at birth with psychiatric conditions such as autism and schizophrenia. The epidemiology for degenerative diseases as diverse as Alzheimer’s disease, Parkinson’s disease and multiple sclerosis all continue to indicate a role of optimal vitamin D status throughout life. This raises the possibility of simple dietary supplementation as an adjunct to current therapies. Although not practicably possible for developmental conditions with adult onset such as schizophrenia, this could eventually be considered in early onset psychiatric disorders, such as autism or ADHD, or for maintaining neural integrity in degenerative conditions via large placebo-controlled, randomised clinical supplementation trials.

We would also like to take this opportunity to highlight design issues in many observational studies opportunistically linking early-life vitamin D deficiency with psychiatric disorders. Most of the published observational epidemiological studies linking early-life vitamin D status with a psychiatric diagnosis never address reverse causality (the condition changes behaviours that lead to less sun exposure). By way of illustration, a very high-profile recent report in the New England Journal of Medicine showed all mental illnesses were associated with an increased risk in a general medical condition [189]. In other words, patients with psychiatric conditions are generally suffering from other conditions that will curtail behaviour perhaps altering diet, exercise and exposure to sunshine. We urge all future epidemiological studies that seek to examine the relationship between vitamin D and psychiatric or neurological conditions to rigorously control for the often-poor general health of patients. This, of course, is less of an issue for gestational (DVD-deficiency) exposures in otherwise healthy mothers.

Important data have emerged from a recent mendelian randomisation study examining gene pathways related to 25(OH)D_3_ blood concentrations. This study found no evidence that genetic factors involved in the synthesis of 25(OH)D_3_ were causal for psychiatric disorders [190]. We have interpreted this to mean any link between 25(OH)D_3_ levels and brain-related outcomes are likely to be solely driven by environmental factors.

Autism is perhaps the developmental brain disorder most regularly linked with DVD-deficiency. However, well-conducted studies refuting this link are now emerging. For the studies describing an inverse relationship between maternal vitamin D levels and autism, they all had mean 25(OH)D_3_ levels of <50 nM which is considered by some authors to represent a cut off for vitamin D deficiency [11,12,13,191]. Two recent studies have failed to find this inverse association [106,192]. So, at face value, this appears a failure to replicate previous studies. However, it is crucial to note that in these last two studies, the mean levels of 25(OH)D_3_ were actually very high (>70–80 nM) and there were very few individuals that were actually vitamin D-deficient, meaning the association could not be properly tested. These same six studies all used the same laboratory to analyse samples, so technical bias (so common amongst vitamin D studies in different populations) could be ruled out. This suggests a threshold effect rather than any continuous relationship between DVD-deficiency and autism. We highlight this particular relationship to illustrate some of the confusion regarding statements regarding potential causality between vitamin D and various brain-related clinical disorders.

The incidence of hypovitaminosis D in both pregnant women and their newborns and the general population remains concerning [193]. Clearly, more rigorous study design is required taking into consideration such issues to bring clarity to the future epidemiological studies. Better quality studies in the future are needed given the substantial emotional and financial burden psychiatric and neurological disorders place on the patient and community. The opportunity to use such a simple, safe and inexpensive intervention as vitamin D supplementation as treatment or as an adjunct to existing therapies in such disorders remains extremely attractive from a public health perspective.

## Figures and Tables

**Figure 1 nutrients-14-04353-f001:**
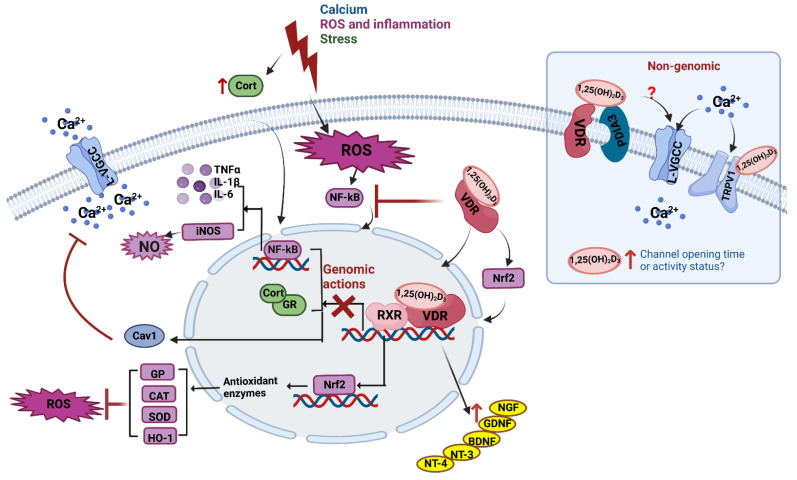
Neuroprotective actions of vitamin D in brain.

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
