# Peer review of "Vitamin D and the Central Nervous System: Causative and Preventative Mechanisms in Brain Disorders"

_nutrients, 2022, doi:10.3390/nu14204353_

Round 1

Reviewer 1 Report

The authors argue that vitamin D plays a vital role in brain development. In particular, they point out that this vitamin mediates increases in neurite outgrowth (p. 30), in neurotrophic factors (p. 4), in TH mRNA (p. 4), in ROS (p. 7), in SOD activity (p. 9), and in serotonin synthesis (p. 9). Development, however, is not simply a matter of growth; it is also a matter of growth cessation. If growth is stimulated beyond the needs of development, the result will be a tumor, perhaps a cancerous tumor.

You can get too much of a “good” thing.” This is especially true for vitamin D. Because it is fat-soluble, excess amounts cannot be excreted in urine and thus build up in the body. Furthermore, optimal serum levels fall within a narrow range: essentially 40 nmol/L to 100 nmol/L in light-skinned humans of the temperate zone. Levels outside that range are associated with higher mortality. In their review of the literature, Tuohimaa et al. (2009) conclude: “There is accumulating evidence that both high and low serum calcidiol concentrations are associated with an increased risk of chronic diseases.” With respect to the central nervous system, the increase in risk with higher vitamin D levels has been shown by a recent murine study:

Increased use of vitamin-D (vit-D) supplements has been attributed for improved cognitive performance, an important consideration given that vit-D deficiency becomes more common with older age. However, several lines of evidence suggest that chronically heightened plasma vit-D may paradoxically compromise cognitive function. The mechanism(s) for detrimental effects of exaggerated vit-D on central nervous system (CNS) function have not been delineated.

… In this study, we present studies in SAMP8 male mice and show that serum 25(OF)D progressively increases with ageing in SAMP8 male. However, we also provide evidence that the increase in serum 25(OH)D occurs concomitant with poorer cognitive performance by MWM analysis and increased capillary permeability. (Lam et al. 2017)

While the authors speak about the benefits of vitamin D, they never discuss the risks. They are thus helping promote the widespread belief in vitamin D as a panacea. The result is a sharp increase in reports of hypervitaminosis D.

Other criticisms

The paper is awkwardly written, and many sentences defy comprehension. The following are a number of examples that caught my eye, but there are many more:

p. 1, lines 31-34

Replace with “Over the last two decades, research on neurons and non-neuronal brain cells, in both human and animal brains, has confirmed not only that the brain possesses the molecular machinery for vitamin D’s actions but also that VDRs are functional, that liganded VDRs directly regulate the expression of target genes and that vitamin D in the form of 1,25-hydroxyvitamin D3 can rapidly alter ion channel function.”

p. 2, lines 78-79

Part of this sentence seems to be missing (after “both provides absolute identification and”).

p. 3, line 126

Replace “express” with “expresses”

p. 3, line 100

Delete the second “which”

p. 3, line 129

Replace “appears not yet been” with “appears to have not yet been”

p. 4, line 191

Replace “1st” with “first”

The authors should reread their paper carefully and make the necessary corrections.

References

Lam, V., Takechi, R. and Mamo, J.C. (2017), [P4–124]: Vitamin D, Cerebrocapillary Integrity and Cognition in Murine Model of Accelerated Ageing. Alzheimer's & Dementia, 13: P1304-P1304. https://doi.org/10.1016/j.jalz.2017.06.1990

Tuohimaa, P.; Keisala, T.; Minasyan, A.; Cachat, J.; Kalueff, A. (2009). Vitamin D, nervous system and aging. Psychoneuroendocrinology 34S, S278-286. https://doi.org/10.1016/j.psyneuen.2009.07.003

Reviewer 2 Report

General comments:

This up-to-date review covering the complex actions of vitamin D and metabolism in the brain is highly appreciated. The authors have compiled and summarized and an ambitious number of papers on the topic. Also the critical comments and words of caution for some types of clinical and preclinical studies seem appropriate and are certainly valuable for non-expert readers.

Specific comments:

1. The Abstract is rather vague and could be modified to better describe what the review contains.  

2. In the Introduction (line 21), the first sentence should be rephrased to avoid misunderstanding; the vitamin D receptor – not vitamin D – is part of the nuclear receptor family.

3. Section III, line 127-130:  Move reference #44 to the previous sentence, which states results from the original study.  Also, in line 129: change “appears not yet been” to apparently has not yet been.

4. Section III B, line 159-161:  Please, rephrase the statement that GDNF and BDNF are “specific to dopaminergic neurons”; also other neuronal populations are responsive to these neurotrophins.

5. Section IV, line 228-231:  Please, rephrase the last sentence of the paragraph – or split into two – for easier comprehension. Also, specify “silencing of the enzyme” (gene silencing by hypermethylation).

6. Section V, line 338:  Please, rephrase “were the possible protective mechanisms [139] “ to “were considered as the possible protective mechanisms [139] “ or similar.

7. Section V, line 264:  Change (TRL4) to (TLR4).

8. Section V, line 382-383:  The full names can be deleted (for smoother reading), the protein symbols have already been introduced. Also applicable at a few other places in the manuscript.

9. Section V, line 421-423:  Is the statement here based results from the previous reference [158]? If not, add the reference.

10. Section V, line 425:  Change the word “myeloid” to myelin. (In the model, myelin proteins are used to induce EAE; myeloid refers to bone marrow-derived blood cells).

11.  The figure is great and helpful but is currently placed at the end of the manuscript, after the Conclusions, where it is not mentioned. It would be better suited embedded in Chapter V, where the illustrated mechanisms are discussed. If this was planned, refer to the figure at the relevant sites of the main text.  A few things seem overly simplified like the DNA-bound VDR, why not draw it as a heterodimer with RXR? Also for VDR at the plasma membrane, would not binding partners like PDIA3 be appropriate?

12. The literature review is extensive and up-to date. Just, from the personal perspective of the reviewer, demyelinating diseases seem to be a bit neglected. Multiple sclerosis is barely mentioned and only in the context of inflammation and ROS, rather than considering direct effects on myelin or oligodendrocytes. Also for some leukodystrophies (or animal models thereof), most of which are characterized by non-inflammatory destruction of brain white matter, effects of vitamin D have been reported.

13.  By and large, the language is adequate and unambiguous. However, careful proof-reading and language/grammar editing is still necessary. For example, a few general points could be improved:

·       -  The word “that” is quite consistently omitted (where needed), in essentially every paragraph of the manuscript.  Also “the” is often missing. 

·       -  Singular and plural verbs are sometimes used incorrectly like, for example, “Our group were”, in the Abstract).

·       -  The proper use of commas and hyphens (throughout the manuscript), would enable more effortless reading.

Some specific language errors and suggested corrections:

·     -    Page 2+5, lines 52, 86 and 240: Change “in respect to" into with respect to or concerning.

·      -  Page 3, line 100: Delete the second “which” (typo).

·    -     Page 3, line 135: “in this space” sounds odd, alternative: in this area or on this topic.

·      -   Page 3, line 142: Change “in line for” to in line with.

·      -   Page 5, line 204: Change “FAC” to flow cytometry.

·      -   Page 6, line 267: Change “trail” to trial (typo).

·      -  Page 6, line 269: Change “does” to dose (typo).

·     -  Page 6, line 289: Change “neurotropin” to neurotrophin (typo).

·      -   Page 8, line 369: Change “transcriptional” to transcription.

·      -   Page 8, line 370-371: Change “properties of antioxidative and antiinflammation” to antioxidative and anti-inflammatory properties.

·      -   Page 10, line 470: Change “improvement” to improved.
